# Microsurgical Reconstruction in Orthopedic Tumor Resections as Part of a Multidisciplinary Surgical Approach for Sarcomas of the Extremities

**DOI:** 10.3390/life12111801

**Published:** 2022-11-07

**Authors:** Georgios Koulaxouzidis, Paul Schlagnitweit, Conrad Anderl, David Braig, Sven Märdian

**Affiliations:** 1Department of Plastic, Aesthetic and Reconstructive Surgery, Congregational Hospital Linz, Sisters of Mercy, 4010 Linz, Austria; 2Department of Orthopaedic Surgery, Congregational Hospital Linz, Sisters of Mercy, 4010 Linz, Austria; 3Department of Plastic and Hand Surgery, Medical Center, Faculty of Medicine, University of Freiburg, 79106 Freiburg, Germany; 4Division of Hand, Plastic and Aesthetic Surgery, Ludwig Maximilian University of Munich, 80336 Munich, Germany; 5Centre for Musculoskeletal Surgery, Charité-University Medicine Berlin, Corporate Member of Freie Universität Berlin and Humboldt-Universität zu Berlin, Berlin Insitute of Health, 13353 Berlin, Germany

**Keywords:** sarcoma, extremity, reconstruction, microsurgery, flap, reconstructive surgery, limb salvage surgery

## Abstract

A central element of modern sarcoma therapy is complete surgical tumor resection with an adequate safety margin, embedded in an interdisciplinary multimodal therapy concept. Along with ensuring patient survival, functional limb preservation is an important goal for sarcomas of the extremities. This review provides an overview of the relevant literature on indications and goals of reconstructive options, the scope and contribution of microsurgical reconstructive procedures, and the associated interdisciplinary decision making and workup. Furthermore, the impact of (neo)-adjuvant therapy on reconstructive decisions will be highlighted. These aspects will be illustrated by four comprehensive case studies that demonstrate both useful strategies and the need for individually tailored therapies. Nowadays, extremity-preserving therapy is possible in more than 90% of sarcomas. Technical and procedural innovations such as microsurgery and microsurgical reconstructive procedures have significantly contributed to this evolution of therapy.

## 1. Interdisciplinary Considerations in Orthopaedic Tumor Resection and Oncologic Reconstruction

Soft tissue sarcomas (STS) are a heterogeneous group of tumors of mesenchymal origin. Currently, more than 70 entities have been described, and the number is increasing. They account for 1% of all adult cancers [1]. Bone sarcomas, also known as primary bone cancer, account for only 0.2% of all malignancies. Approximately 60% of sarcomas are initially diagnosed at a localized stage, and 18% are diagnosed at a locally advanced stage. Sarcomas are predominantly localized at the extremities (60%), 15% at the upper and 30–45% at the lower extremity [2].

The central element of modern guideline-compliant therapy is complete surgical tumor resection with an adequate safety margin. This should be embedded in an interdisciplinary multimodal therapy concept in accordance with the guidelines. Therapy should be accompanied by a specialized interdisciplinary tumor case conference for sarcomas at a comprehensive cancer center [3,4].

As surgical treatment for sarcomas has evolved, so have expectations for outcomes. While for decades amputation was the standard procedure to ensure survival, extremity-preserving therapy is now possible in more than 90% of sarcomas [5,6]. Increasing knowledge of tumor biology has contributed significantly to this development. Remarkable studies have shown that extremity-preserving resection in combination with radiotherapy, when technically feasible, is equivalent to the oncologic safety of amputation [7]. Moreover, both technical and procedural innovations have also contributed to this progress. The former includes state-of-the-art surgical microscopes and corresponding microsurgical instruments, e.g., microsurgery and the associated reconstructive possibilities [8]. The next step in this evolution of techniques is supra-microsurgery, which allows us to perform lymphatic anastomoses for instance. This seems to be an upcoming issue for the prevention of lymphedema–lymphocele after sarcoma resection [9]. Additionally, the growing knowledge of flap physiology has contributed significantly to our capabilities. The remarkable work of G. Ian Taylor et al. on the “Angiosome Concept and Tissue transfer” represents a milestone in reconstructive innovation [10,11]. 

Neoadjuvant and adjuvant therapy usefully complement surgery to increase success rates. Depending on tumor biology, a possible option in localized tumor disease is neoadjuvant therapy to reduce the tumor mass. In cases of success, it might lead to reduced morbidity thanks to resection. For tumors with a high risk of metastasis, neoadjuvant chemotherapy may additionally target potential micrometastases. However, depending on the tumor entity, this strategy is not universally applicable and not consistently successful in these targets. Therefore, a combined reconstructive challenge is often the case. Neoadjuvant radiotherapy causes collateral damage to surrounding tissue and structures. This damage zone extends well beyond the margins of the tumor. In terms of adverse effects, this damage zone is comparable to the “zone of injury” of a trauma situation [12], leading to decreased quality of potential recipient vessels or local reconstruction possibilities and a significant increase in the risk of vascular complications and wound healing disorders [13]. The former is highly the case, as tumor patients already have an increased coagulability and risk of thrombosis. In contrast, several studies have shown that the long-term functional outcome in extremity sarcomas after reconstruction is positively affected by neoadjuvant radiotherapy compared with adjuvant therapy [14]. One reason is a lower radiation dose, as the targeted volume is clearly defined, and thus less adjacent tissue will be irradiated. Furthermore, subsequent reconstruction is performed with unirradiated tissue, while tumor resection involves the removal of significant portions of irradiated tissue. It should be noted that the increased risk and incidence of early postoperative wound healing disorders are inconsistently discussed [15]. Recent studies have shown that the postoperative complication rate for reconstruction with free tissue transfer is comparable after both neoadjuvant and adjuvant radiotherapy [16]. This is in contrast to assumptions made in the past [14]. In the case of locally advanced disease, isolated limb perfusion (ILP) with TNF-a is another option to achieve operability under certain conditions and in individual cases. However, this option is only applicable to individual centers.

The timing of reconstruction depends on patient-related factors, such as resilience for advanced surgery, the certainty of complete tumor resection, and the surgeon’s evaluation. Primary single-stage reconstruction is associated with various advantages [17]. The conditions for reconstruction are more favorable due to less tissue inflammation, edematization, and lack of scarring or fibrosis. The risks of bacterial wound contamination or wound healing disorders are minimized. The number of surgeries required is reduced, and the risk of potential delays of adjuvant therapy is minimized [18]. Nevertheless, the differences regarding the rate of perioperative complications in the comparison of immediate and two-stage reconstruction are controversially discussed. The main indication for delayed reconstruction is an uncertain complete tumor resection. In this case, temporary wound closure is necessary until the resection status will have been clarified. 

In concordance with the advances in soft tissue reconstructions, the techniques of bone defect reconstruction (e.g., bone segment transfers, vascularized bone transfer, etc.) or endoprosthetic replacements have improved considerably. This also shifted the limits of limb-sparing resections to the point that today all major joints can be replaced by modular megaprostheses. Hence, even complex partial pelvic resections can be reconstructed with good to excellent functional results. 

## 2. Goals of Limb Reconstruction

The primary goal of modern curative therapy of soft tissue sarcomas is to ensure survival while avoiding local tumor recurrence or distant metastases. In addition, however, other ambitious purposes are being pursued today. These include:Low risk for postoperative complication, which also allows adjuvant therapy without time delay;Aesthetically satisfying results and rapid social reintegration contribute to a high degree of quality of life.

Along with the above, the primary goal in localized disease is to preserve an extremity that is functionally as unrestricted as possible without compromising oncologic safety [6,8]. The radicality of tumor resection significantly influences this oncological safety. On the other hand, a sensibly coordinated application of the accompanying therapy methods—such as radiotherapy or chemotherapy—is decisive and must be oriented towards guidelines in terms of timing and modality. However, this requires a robust, stable, and resistant reconstruction [19]. Neoadjuvant or adjuvant chemotherapy or radiotherapy compromises wound healing and may also influence the quality and vitality of tissue grafts. Thus, simple skin grafting as a reconstruction procedure rarely meets the above requirements in the context of (neo-)adjuvant radiotherapy [20]. Reconstruction should be associated with a low risk of systemic and local complications. In addition to a wide resection with a sufficient safety margin, oncologic safety and outcome are defined by a synchronized application of a multimodal therapeutic approach without delay due to complications. Depending on the anatomical location, specific requirements for reconstruction exist. An example would be that pain-free reconstruction for the upper and lower extremities is essential [21]. Furthermore, from a functional point of view, stability is particularly crucial for the lower extremity, while free joint and tendon mobility plays a major in the upper extremity [22,23]. Finally, reconstruction should achieve an aesthetically pleasing result if possible. This contributes significantly to the quality of life and social reintegration. 

The entire interdisciplinary armamentarium of reconstructive procedures offers a wide range of possibilities for achieving the goals of meaningful limb reconstruction while taking individual aspects into account. This is exemplified in the first case shown (Figure 1), which is a Leiomyosarcoma of the distal thigh (pT2b (10 cm), N0, M0, G2). A primary wide resection (R0) was performed, including the distal femur and knee. This is followed by immediate endoprosthetic reconstruction and soft tissue reconstruction with a free fasciocutaneous anterolateral thigh flap (ALT flap). The multimodal therapy concept was completed by adjuvant radiotherapy.

## 3. Contraindications to Limb Preservation and Amputation Claims

Numerous parameters influence the decision-making process of limb preservation versus primary amputation: Tumor size, anatomical location, degree of invasion, affected functional relevant anatomical structures, and the estimated functional outcome after resection are crucial factors to consider. Moreover, the patient’s general status and wishes must be considered. A functional outcome that hinders daily activity and a high degree of morbidity argues against a preservation attempt. However, the patient’s age per se is not a counterargument for limb preservation. Nevertheless, the surgical team must ensure that the patient is resilient enough to tolerate prolonged surgery with extensive wound surfaces.

If limb function is lost due to the necessary degree of radicality, reconstruction and functional optimization should be pursued without compromise using the entire spectrum of reconstructive procedures. This also applies to optimizing an amputation stump, e.g., by adjusting the stump length or its soft tissue coverage. For this purpose, bone lengthening measures, such as distraction osteogenesis, as well as free tissue grafts for adequate cushioning of bony pressure zones without excess soft tissue, taken from the amputated limb (spare part surgery), can be useful to enable a stable, functional, and pain-free prosthetic fitting [24]. In particular, the concept of spare part surgery can offer remarkable opportunities for functional as well as aesthetic optimization. The second case presented was a sarcoma (NOS) of the left shoulder (pT2, N0, M0, G3). We performed a wide resection by interscapulothoracic arm amputation (forequarter amputation; R0) and reconstruction of the shoulder silhouette and axilla with a free composite elbow tissue transfer (Figure 2). Hereby a gothic arc hemithorax after amputation of the upper arm could be avoided.

Especially in the lower extremity, the advantages of limb preservation for locally advanced malignancies with the need for extensive reconstruction compared to amputation are controversial. The results of the Lower Extremity Assessment Project (LAEP) study have provided valuable and partially transferable insights into the benefit of limb preservation vs. amputation [25,26]. However, these results were obtained on traumatic high-energy lower extremity injuries. Therefore, transmission to the case of a localized malignancy is limited. Nevertheless, conclusions for the elective and planned case of malignant events of the lower extremity also arise here. A non-sensitive foot per se, for example, does not discriminate against limb preservation or primary amputation [26]. A painless result is a critical aspect in the context of both limb-preserving procedures and amputations. The concept of “targeted muscle reinnervation” described by Souza et al. contributes to reducing the risk of future neuroma-related pain and thus to a better outcome [27]. 

## 4. Palliative Plastic Surgery in Extended Sarcoma Disease

With the possibilities of plastic surgery, an improvement in the quality of life for patients in a palliative situation can be achieved individually within the framework of the multimodal therapy concept. This happens especially when conservative therapies, such as wound treatment, fail to achieve manageable conditions. For example, exulcerated tumors can be resected and an improved nursing situation can be achieved with low-risk reconstructions. Although rare, complicated wounds following radiation therapy may represent another indication. In order to bring these therapy options to the patients, plastic surgeons have to attend relevant oncological conferences [28].

## 5. Microsurgical Reconstruction

Due to the limited local soft tissue available, the likelihood of microsurgical reconstruction procedures such as free flap reconstruction, nerve, or vascular reconstruction is higher on the distal half of the extremities than on the proximal half or trunk [29]. However, if vascular reconstruction is required, physicians should be aware of the increased perioperative complication rate. Similarly, the chance of microsurgical reconstruction procedures is higher in the upper extremity than in the lower [30]. Numerous studies have reported improved functional and aesthetic outcomes with microsurgical reconstructive procedures [8,30]. This is illustrated by the third case presented (Figure 3). The case shows a synovial sarcoma of the dorsum of the foot (pT1a, N0, M0, G2,) after incomplete resection during a “whoops” procedure. We performed a wide resection of the dorsum of the foot including tendons, ligaments, and partial muscles. During reconstruction, the extensor tendons were fixed bony and the soft tissue defect was reconstructed by a free fasciocutaneous anterolateral thigh flap (ALT flap). Due to its nonspecific clinical presentation, ubiquitous occurrence, and rarity, “whoops” surgery in the case of sarcomas is a common occurrence. A prognosis is not affected negatively by immediate wide resection in the context of multimodal therapy. 

Local or regional flap reconstruction is often associated with a higher complication rate and substandard functional outcomes. This results from additional surgical trauma accompanied by a vascular and lymphatic comprise of the affected limb. Free tissue grafts can provide large volumes of well-perfused tissue and thus avoid dead spaces. Apart from that, they are resistant to radiotherapy due to their vascularity or, when used after neoadjuvant radiotherapy, can contribute to stable wound healing. Furthermore, the use of free tissue grafts allows for uncompromised radical resection. Next to free muscular grafts with split-thickness or full-thickness skin grafting-various fasciocutaneous and osteocutaneous flaps represent the workhorses of extremity reconstruction [31]. 

Muscle flaps usually atrophy in case of denervation. They also adhere particularly well to the wound bed and are less mobile compared to fasciocutaneous flaps. In our experience, these properties are advantageous, for example, in reconstructing weight-bearing lower extremities or the palm [32]. However, this is controversially discussed in the literature [33].

## 6. Upper Extremity

Function is the primary goal of reconstruction in the upper extremity. Nevertheless, an aesthetic result is also advantageous as the upper extremity is often exposed. Owing to the close anatomical relationship of functional structures, such as the neurovascular bundle or tendomuscular units, their preservation without comprising radicality or inclusion in resection is regularly questioned, especially in high-grade tumors or advanced disease [23]. In addition to the need for defect coverage, functional reconstructions with motor replacement procedures, nerve and/or vascular grafts are often indicated.

The team and the unit in charge of the reconstruction must apply the entire spectrum of reconstructive procedures in terms of skills and infrastructure to avoid compromises in radicality and enhance functional outcomes. Motor replacement operations include tendon or nerve transfers, free or pedicled functional muscle flaps (e.g., gracilis muscle, rectus femoris muscle, or latissimus dorsi muscle) or nerve grafts (e.g., sural nerve). Free tissue grafts as well as local or regional flaps can be used for less demanding reconstructions. The workhorse for extensive defects is the latissimus dorsi muscle flap. Besides the possibility to cover even significant defects, it might even be used as an innervated version for functional reconstruction of elbow joint flexion or extension following extensive resection. Alternatively, the vastus lateral muscle, for example, can be used for extended defects in the supine position with a comparable pedicle length. However, perforator-based flaps, such as the thoracodorsal artery perforator flap (TDAP) or perforator flaps based on the circumflex scapular arterial system (e.g., parascapular flap) expand our options and contribute to a reduction in donor site morbidity. If a fasciocutaneous flap is required, the anterolateral thigh (ALT) flap [34], the parascapular flap, or the thoracodorsal artery perforator (TDAP) flap represent the most commonly used variants.

The characteristics of the mainly used flaps are summarized in Table 1.

Meanwhile, most free muscle flaps, fasciocutaneous flaps, or perforator flaps have low to acceptable donor site morbidity. In case of a proximal localization of the extremity tumor conventional random pattern flaps or axial pattern flaps from the upper extremity (e.g., lateral upper arm flap) or shoulder region can often be used successfully. Axially perfused flaps, such as the dorsal interosseous flap, are an option for reconstructions of the forearm or hand. The radial and ulnar arteries, which represent the main supply vessels of the forearm and hand, should only be sacrificed for local pedicled flap reconstruction in exceptional cases due to the increased donor site morbidity associated with it. For the reconstruction of extensive bony shaft defects, free vascularized bone grafts such as the fibular graft may be a reliable choice [31]. Tumor endoprosthetic reconstructions are an option in cases of affections of joint structures or structures close to the joint [35]. 

## 7. Lower Extremity

The primary reconstructive purpose in the lower extremity is not only the function but predominantly the stability to ensure sufficient ambulation. Overall, the indication for free microsurgical procedures for reconstruction is more frequent in the distal half of the lower extremity. Especially in the adductor region, primary direct closure of the defect is usually possible. 

The gluteal region can be challenging. Indicated wider resections that include abducting muscles, such as the gluteus medius and/or minimus muscle, are rare. However, they result in a devastating functional impairment, such as the Trendelenburg deformity. Transfer of the gluteus maximus muscle and tensor fascia lata for primary deficiency of the abductors of the hip, as described by Whiteside et al., may be a valid functional reconstructive option [36].

Most bone or joint defects at the lower extremity may be reconstructed using modular tumor prosthetic systems and alloplastic or heteroplastic bone replacement. These procedures are more common in the lower extremity compared to the upper. However, the autologous fibula flap for bone defects, whether pedicled (e.g., fibula pro tibia procedure) or freely transplanted, is an established and safe procedure. In endoprosthetic joint replacements and osteoarticular allografts, the failure rate can be as high as 40% after ten years [37]. In recent years, this high complication rate has been reduced due to innovations in implant technology. Coated tumor prostheses—mostly with silver—have been introduced and led to a reduction in infection and revision rates [38]. Additionally, the risk of delayed wound healing and secondary infections is significantly increased in the lower extremities with enormous consequences. Therefore, a primary competitive soft tissue optimization or reconstruction with free, well-vascularized tissue flaps is often indicated to reduce the risk of complications associated with the above bone replacement procedures. The criteria for flap selection as well as the recommendations for the flaps to be selected are consistent with those previously mentioned. 

Loss of foot elevator function can be reconstructed by tendon transfer techniques, neuromusculotendinous transfer [39], orthosis, or targeted nerve stimulation by external or implantable pacemakers (Figure 4). Worth mentioning is the remarkable functional results of a transtibial amputation with adequate prosthetic fitting. They make heroic complex interdisciplinary interventions with expected moderate functional results in advanced localized disease questionable [5,26,40].

## 8. Summary

Radical tumor resection combined with potent reconstructive procedures provides a treatment that makes virtually no compromises in oncologic safety while ensuring a functional, aesthetically pleasing, and sustainable result. This further expands the options for neo- or adjuvant therapies. In single cases, isolated limb perfusion allows an attempt at curative limb preservation even in the presence of highly advanced local disease. Microsurgical reconstruction is an essential part of tumor surgery. It provides the orthopedic tumor surgeon with the prerequisites for uncompromising radical resection. For the reconstructive surgeon, it enables the provision of reliable, well-vascularized tissue for defect coverage as well as for functional reconstruction even in complex situations. In this review, we present current principles of reconstruction in orthopedic tumor surgery (Figure 5). The following questions are asked for orientation:Can the resulting defect be closed primarily without tension?Do additional risk factors for wound healing delays or complications exist?

Recommendations for the definition of a defect suitable for simple skin-graft reconstruction are summarized as well as risk factors for a higher likelihood of wound healing disorders in the case of primary closure. Some of these patient-specific risk factors may include a past or planned radiotherapy, compromised vascular status of the extremity, obesity, nicotine abuse, or poorly controlled diabetes mellitus.

What is the primal purpose of reconstruction?
○Defect coverage or dead space obliteration (A), and/or ○Functional reconstruction (B)


Several opportunities are presented for the specific goal of reconstruction. 

Is the defect proximal or distal to the limb?

Under specific circumstances proximally located defects can often be reconstructed by locoregional procedures 

Where is the defect mainly located?
○In a functional dynamic area, such as joints, or○In a static diaphyseal area


The functional area is defined as an area at the joints (e.g., elbow, knee) or where tendons glide extensively (e.g., dorsum of the hand or distal forearm). A static area is mainly the area at the diaphysis of the long bones, which is mainly characterized by muscles (e.g., upper arm or thigh). 

Finally, we present our preferred reconstructive options based on location and primary reconstruction goal. However, there are a number of other reliable options. 

In Table 1 we summarized the main characteristics of the mainly used free flaps for reconstruction.

Critical indication and interdisciplinary coordination as well as the possibility and willingness to apply all available reconstructive procedures are the basis for a sustainable, functional, and satisfactory result for both the practitioner and, in particular, the patient. 

## Figures and Tables

**Figure 1 life-12-01801-f001:**
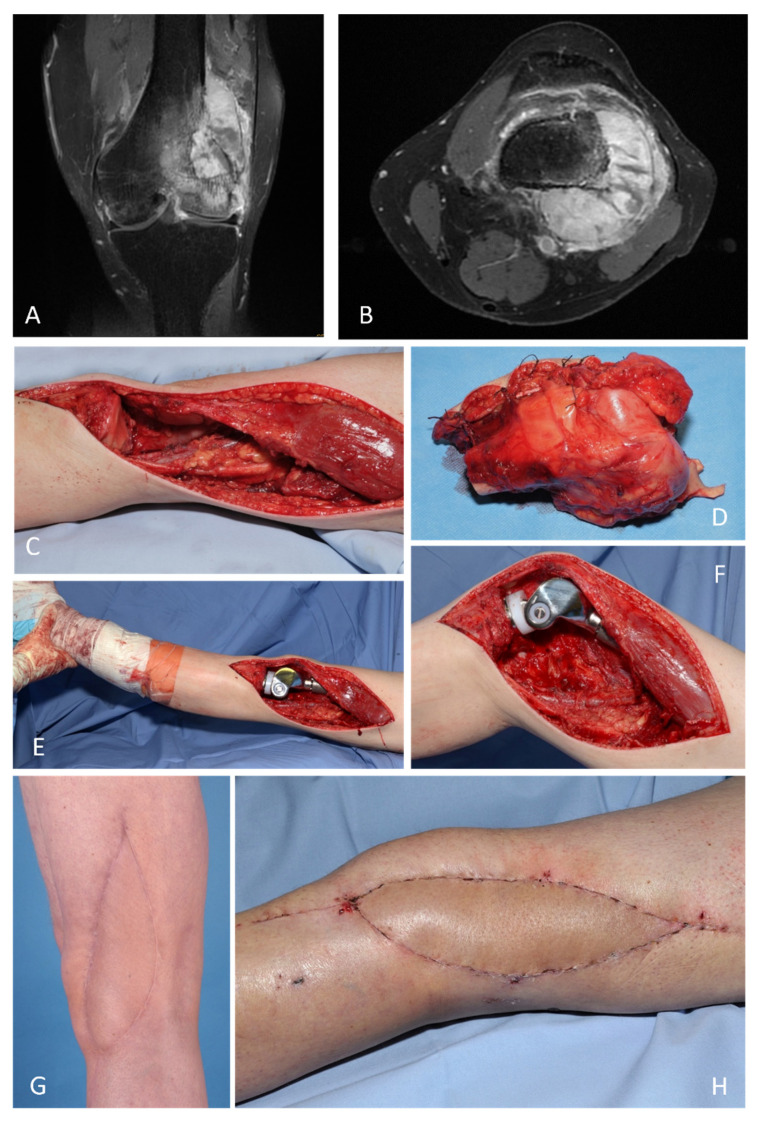
Knee joint reconstruction with interdisciplinary reconstructive methods. Leiomyosarcoma of the distal thigh (pT2b (10 cm), N0, M0, G2). Primary wide resection (R0) with interdisciplinary reconstruction of the knee, distal femur and soft tissues. Adjuvant radiotherapy: (**A**,**B**) MRI of the knee with demonstration of tumor. (**C**) Surgical site of the knee and distal thigh after tumor resection. (**D**) Resected tumor. (**E**,**F**) Overview and close-up look after endoprosthetic knee and femur reconstruction. (**G**,**H**) Overview and close-up after soft tissue reconstruction with a free fasciocutaneous anterolateral thigh (ALT) flap (ALT-flap).

**Figure 2 life-12-01801-f002:**
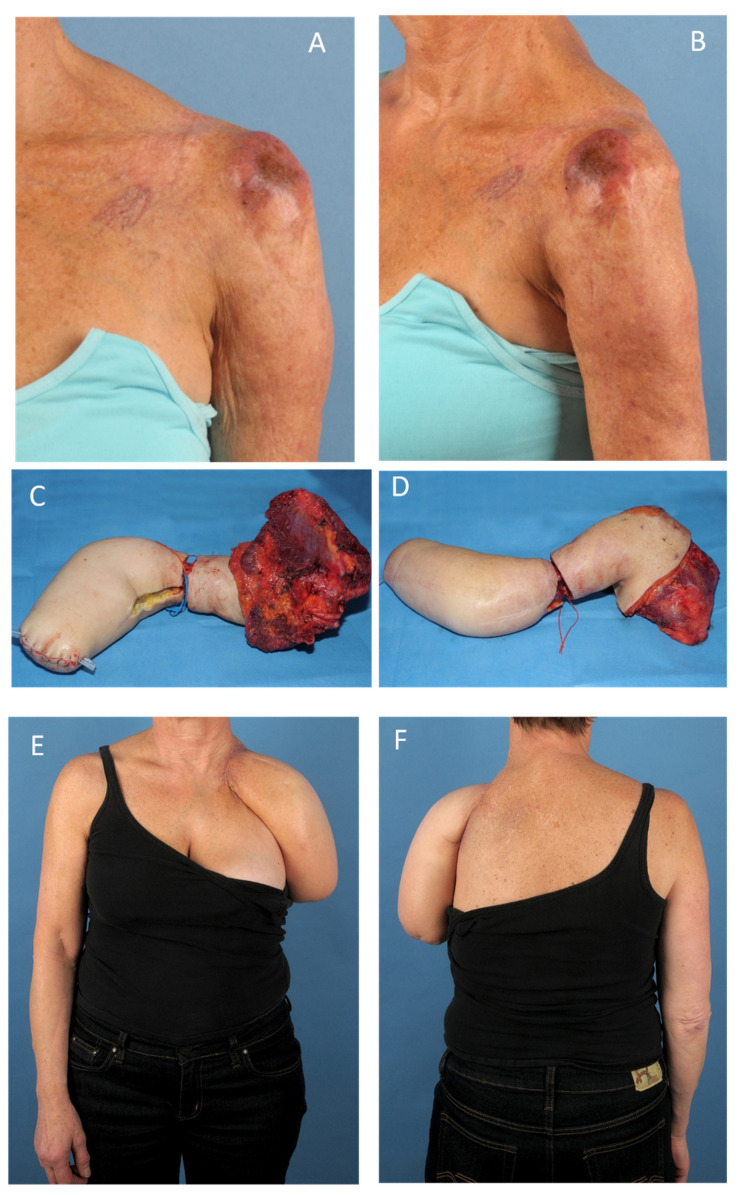
Spare part surgery for an optimized result after forequarter amputation. Sarcoma (NOS) of the left shoulder (pT2, N0, M0, G3). Primary wide resection by interscapulothoracic arm amputation (R0) and reconstruction of the shoulder silhouette and axilla with a free composite elbow tissue transfer: (**A**,**B**) Sarcoma in the left shoulder. (**C**,**D**) Widely resected shoulder tumor after interscapulothoracic arm amputation, with in situ prepared autologous elbow graft. (**E**,**F**) Reconstructed shoulder silhouette.

**Figure 3 life-12-01801-f003:**
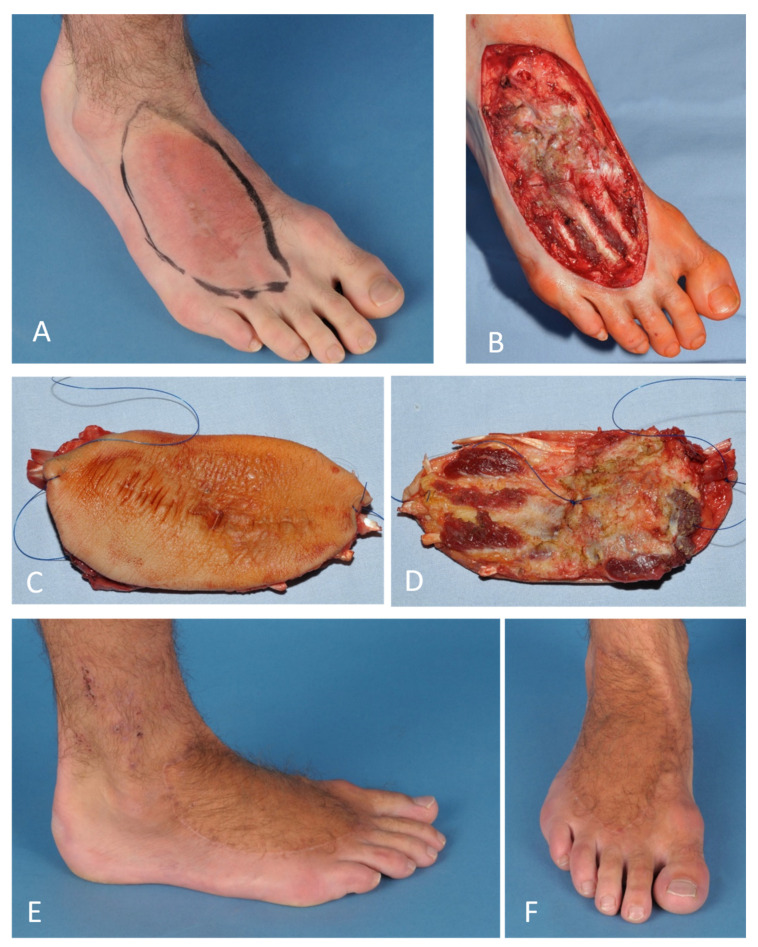
Improved functional and aesthetic outcomes through microsurgical reconstructive procedures: (**A**) Synovial sarcoma of dorsum of the foot (pT1a, N0, M0, G2,) after incomplete resection. (**B**) Dorsum of the foot after wide resection. (**C**,**D**) Wide resection of the tumor including tendons, ligaments, and partly muscles. (**E**,**F**) Reconstruction with osseous fixation of the extensor tendons and by a free fasciocutaneous anterolateral thigh flap (ALT flap).

**Figure 4 life-12-01801-f004:**
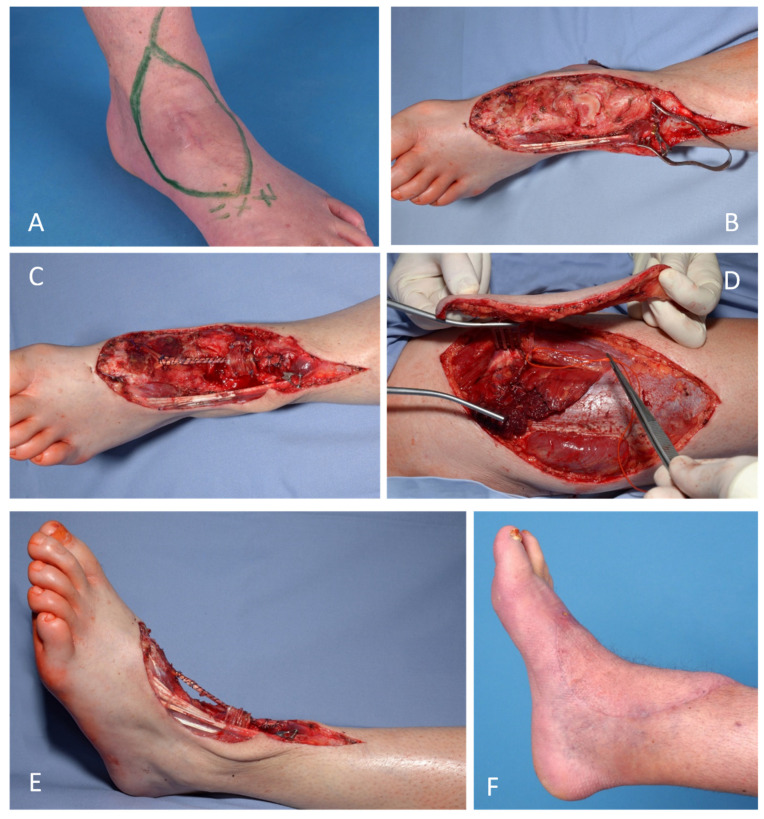
Combined procedures for an optimal oncological and functional end result: (**A**) Myxofibrosarcoma of the dorsum of the foot presented after incomplete resection (R2) (cT1, N0, M0). (**B**) Major radical tumor resection (R0). (**C**–**E**) Primary reconstruction of the tendon of the tibialis anterior muscle by a fascia lata graft. (**D**) Soft tissue reconstruction by a free fasciocutaneous anterolateral thigh flap (ALT flap). (**F**) Late functional, stable and aesthetic reconstructive final.

**Figure 5 life-12-01801-f005:**
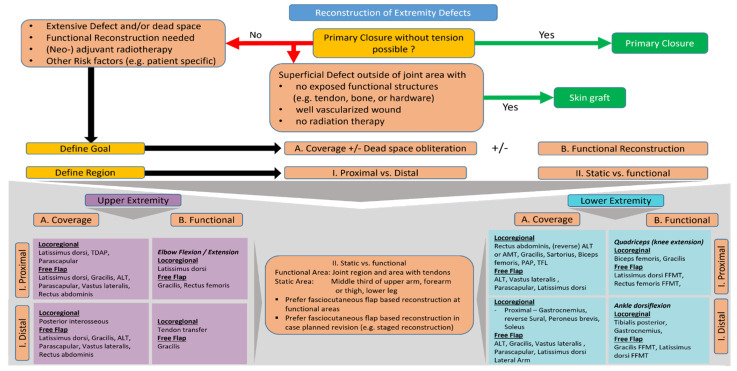
Algorithm for reconstructive decision as part of a multidisciplinary surgical approach for sarcomas of the extremities. Summary of current principles in reconstructive decision making and planning in the context of orthopedic tumor resections as part of a multidisciplinary surgical approach for sarcomas of the extremities. TDAP—Thoracodorsal artery perforator flap; ALT—anterolateral thigh flap; AMT—Anteromedial thigh flap; PAP—Profunda artery perforator flap; TFL—tensor fascia lata flap; FFMT—free functional muscle transfer.

**Table 1 life-12-01801-t001:** Workhorses of reconstructive flaps and their characteristics.

Flap	Flap Type	Dominant Pedicle Vessels	Pedicle Length	Vascular diameter
Latissimus dorsi flap (LD)	MF, MCF, FFMF	TDA	8 cm	2.5 mm
Thoracodorsal artery perforator flap (TDAP)	FC	TDA	8 cm	2.5 mm
Parascapular flap	FC	CSA	5–6 cm	2.5 mm
Anterolateral thigh flap (ALT)	FC, MFC	Descending branch LCFA	12 cm	2 mm
Vastus lateralis musculus quadriceps femoris flap	MF	Descending branch LCFA	12 cm	2 mm
Anteromedial thigh flap (AMT)	FC, MFC	Descending branch LCFA	12 cm	2 mm
Tensor fascia lata flap (TFL)	MF, MFC	Ascending branch LCFA	7 cm	2–3 mm
Profunda femoris artery perforator flap (PAP)	FC, MFC	PAP	10 cm	1.5–2 mm
M. gracilis	MF, MFC	Ascending branch LCFA	6 cm	1.6 mm

MF: Muscle flap; MCF: Musculocutaneous flap; FFMF; Free functional muscle flap; FC: Fasciocutaneous flap; TDA: Thoracodorsal artery and venae comitantes; CSA: Circumflex scapular artery and venae comitantes; LCFA: lateral circumflex femoral artery and venae comitantes; PAP: Profunda femoris artery perforator.

## Data Availability

We exclude this statement because no data are reported in this review.

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
