# Peer review of "Microsurgical Reconstruction in Orthopedic Tumor Resections as Part of a Multidisciplinary Surgical Approach for Sarcomas of the Extremities"

_life, 2022, doi:10.3390/life12111801_

Round 1

Reviewer 1 Report

-        Between page 1 and 2, the authors wrote: Therapy should be guided by an interdisciplinary tumor conference at a comprehensive cancer center. Is this sentence correct?

-        In lines 61-62 the authors wrote:  As a result of this development, thereconstructive ladder is now used 61 as a reconstructive -elevator depending on local capabilities and the experience of the re-62 constructive team. I believe there is an error in this sentence.

-        Figures are labeled 2 times

-        Several references throughout the manuscript do not appear.

-        The authors must improve the last section of the manuscript, summary. It is one of the most relevant parts of the manuscript.

-        All manuscript must be revised for grammatical errors.

-        To be a review, the number of references is limited.

Author Response

Dear Ladies and Gentlemen,

Many thanks for the constructive and helpful comments. In our estimation, this has contributed to a significant improvement and structuring of the manuscript and we hope that this will be recognized accordingly by the reviewers and the Editor in Chief. All changed areas have been marked in the document.

Kind regards

Georgios Koulaxouzidis

Reviewer 1 (Round 1)

  • Between page 1 and 2, the authors wrote: Therapy should be guided by an interdisciplinary tumor conference at a comprehensive cancer center. Is this sentence correct?

Thank you for this important note, we have corrected the sentence.

  • In lines the authors wrote: As a result of this development, the reconstructive ladder is now used as a reconstructive -elevator depending on local capabilities and the experience of the reconstructive team. I believe there is an error in this sentence.

Thank you for this important note, we have corrected the sentence.

  • Figures are labeled 2 times

Thank you for this important note, we have adjusted the designations.

  • Several references throughout the manuscript do not appear.

We have checked and corrected this. In addition, we have removed some redundant references.

  • The authors must improve the last section of the manuscript, summary. It is one of the most relevant parts of the manuscript.

Thank you for this important note regarding the relevance of the summary. We have condensed the structure by providing detailed explanations of the decision criteria and thus the comprehensibility of the algorithm. Furthermore, we believe that it is valuable to present the most commonly used free tissue grafts (free flaps) with their main characteristics in a tabular summary, even though there is quite a lot of literature on this subject.

  • All manuscript must be revised for grammatical errors.

We have revised the complete manuscript both linguistically and grammatically, eliminating errors and also significantly improving readability.

  • To be a review, the number of references is limited.

Thank you, taking into account this valid comment, the number of references has been reduced.

Reviewer 2 Report

Thank you for submitting a  comprehensive review.

Author Response

We would like to express our sincere gratitude for the favorable and positive assessment of the quality and suitability of our work for publication.

English language and style as well as orthography are now improved. 

Yours sincerely, 

Koulaxouzidis Georgios 

Reviewer 3 Report

The article summarizes the main aspects of the management of sarcomas of the extremities, with a focus on plastic reconstruction.

The management of sarcomas should be limited to the few centers that can guarantee an integrated multidisciplinary approach (oncologist, orthopedic surgeon, radiotherapist, dedicated radiologists, plastic surgeons, vascular surgeons, etc.). In fact, the rarity of these pathologies and the complexity of management (considering the many possible variables) make it difficult to achieve mastery of the topic. I think that articles of this type, with a "didactic" approach may still be useful in the literature precisely considering these aspects. However, this article proceeds in my opinion in a rather disorganized manner; it is difficult to follow the discussion even for a reader familiar with the topic. The aspects covered are poorly explored, especially the main topic, namely microsurgery.

I think the article can be considered for publication only after a major rearrangement.

1) the steps of sarcoma management must be clearly separated, with attention paid to describing the aspects in the cases of localized tumor, locally advanced tumor, and metastatic tumor;

2) it is necessary to clearly divide the considerations regarding sarcomas of bone and those regarding soft tissue sarcomas:

3) although it is not the main topic of the article, more space should be dedicated in my opinion to the description of orthopedic reconstruction techniques in case of bone tumors, this being the central part of reconstructive treatment;

4) focusing on the plastic surgery aspects, I think it is necessary to provide a table describing individually the various possibilities of fasciocutaneous and muscle flaps with pros and cons; this is also in order to make the interesting proposed concluding scheme more understandable.

Author Response

Dear Ladies and Gentlemen,

Many thanks for the constructive and helpful comments. In our estimation, this has contributed to a significant improvement and structuring of the manuscript and we hope that this will be recognized accordingly by the reviewers and the Editor in Chief. All changed areas have been marked in the document.

Kind regards

Georgios Koulaxouzidis

Reviewer 3 (Round 1)

The article summarizes the main aspects of the management of sarcomas of the extremities, with a focus on plastic reconstruction.

The management of sarcomas should be limited to the few centers that can guarantee an integrated multidisciplinary approach (oncologist, orthopedic surgeon, radiotherapist, dedicated radiologists, plastic surgeons, vascular surgeons, etc.). In fact, the rarity of these pathologies and the complexity of management (considering the many possible variables) make it difficult to achieve mastery of the topic. I think that articles of this type, with a "didactic" approach may still be useful in the literature precisely considering these aspects. However, this article proceeds in my opinion in a rather disorganized manner; it is difficult to follow the discussion even for a reader familiar with the topic. The aspects covered are poorly explored, especially the main topic, namely microsurgery.

I think the article can be considered for publication only after a major rearrangement.

1) the steps of sarcoma management must be clearly separated, with attention paid to describing the aspects in the cases of localized tumor, locally advanced tumor, and metastatic tumor;

Thank you very much for the comment. The paper focuses on microsurgical reconstructive options. It places them in the interdisciplinary context of a multimodal sarcoma therapy. In doing so, an attempted focus is left on reconstruction. A detailed presentation of the reconstruction options depending on the stage of the disease is only possible to a limited extent due to the many influencing factors in the specific decision. Nevertheless, we have taken the advice and improved the structure of the manuscript by presenting general aspects such as the interaction of concomitant therapeutic modalities on reconstruction in a concentrated manner in the first section of the manuscript. In addition, these are cited in a bundled manner for the respective concomitant modalities. Thus, the sections on the lower and upper extremities concentrate on the specific principles of each extremity.

We hope that this limited consideration of the reviewers advice is acceptable.

2) it is necessary to clearly divide the considerations regarding sarcomas of bone and those regarding soft tissue sarcomas:

Thank you very much for this comment. We can understand the request for an equally detailed presentation of the approach to bony sarcomas. However, bone sarcomas and tumor orthopedic reconstruction are not the subject of this paper. We assume that a corresponding article focusing on bone sarcomas and tumor orthopedic reconstruction will appear in this special edition, in which this topics will also find its place.

3) although it is not the main topic of the article, more space should be dedicated in my opinion to the description of orthopedic reconstruction techniques in case of bone tumors, this being the central part of reconstructive treatment;

We refer to our commentary under point 2. For the same reasons, in our opinion, a dedicated presentation of orthopedic reconstructive procedures is not useful in this paper and does not represent the clinical focus of the main author.

4) focusing on the plastic surgery aspects, I think it is necessary to provide a table describing individually the various possibilities of fasciocutaneous and muscle flaps with pros and cons; this is also in order to make the interesting proposed concluding scheme more understandable.

Thank you very much for this extremely helpful advice, which we have taken into account. Accordingly, the characteristics and properties of the most frequently used workhorses of microsurgical reconstruction have been presented in tabular form. However, the advantages and disadvantages of the respective reconstruction method are always to be assessed depending on the individual and concrete situation and can therefore not be presented with corresponding bindingness in this rather general framework. Nevertheless, the principle advances of fasciocutaneous flaps in dynamic areas as well as in case of planed re-operation are highlighted in the manuscript.

Yours sincerely,

Georgios Koulaxouzidis

Round 2

Reviewer 3 Report

The authors have addressed all my concerns. Thank you.